# Effects of Long-Term Storage on Hatchability and Incubation Length of Game Farmed Quail Eggs

**DOI:** 10.3390/ani13132184

**Published:** 2023-07-03

**Authors:** Pedro González-Redondo, Pascual Robustillo, Francisco P. Caravaca

**Affiliations:** Departamento de Agronomía, Universidad de Sevilla, Carretera de Utrera km 1, 41013 Seville, Spain; pedro@us.es (P.G.-R.); pascualrobus@hotmail.com (P.R.)

**Keywords:** quail, game farming, hatching egg, storage, incubation

## Abstract

**Simple Summary:**

In alternative poultry production systems, a key aspect for the viability of the hatching egg is its storage before incubation, its management being less standardized than in intensive poultry farming. The objective of this work is to investigate the effect of the long-term storage of game farmed quail eggs by testing storage periods of up to 35 days at 15.8 °C and 80% relative humidity. It was found that the hatchability was maintained in eggs stored up to 28 days and decreased when storage was extended to 35 days. The eggs progressively lost more weight as the storage time increased, producing chicks with lower weight from eggs stored for more than 14 days and with lower relative chick weight in those stored for 35 days. The length of the incubation period increased progressively with storage time, while hatching synchrony decreased. In conclusion, if necessary for management reasons, game quail hatching eggs can be stored without relevant loss of viability for up to 28 days before incubation. The practical implications are that this ensures a long market life when hatching eggs are sold and allows small farms to collect enough eggs over an extended period to fully set an incubator.

**Abstract:**

The long-term storage of eggs before incubation is a common practice in some alternative poultry systems but needs to be performed under conditions that preserve egg viability. The effects of the long-term storage of game farmed quail (*Coturnix coturnix*) eggs on weight loss during the storage and incubation periods, chick weight at hatch, hatchability, and incubation length were investigated. The eggs were arranged in six treatments submitted to 0-, 7-, 14-, 21-, 28-, and 35-day storage periods at 15.8 °C and 80% relative humidity. The storage length reduced the hatchability of eggs (*p* < 0.05) when the storage was extended to 35 days, decreasing by more than half compared to eggs stored up to 28 days. Egg weight loss during storage progressively increased with the storage length (*p* < 0.05). Chick weight at hatching was reduced in eggs stored for more than 14 days (*p* < 0.05), and relative chick weight decreased significantly in eggs stored for 35 days (*p* < 0.05). Incubation length progressively increased with the storage length (*p* < 0.05), achieving less hatching synchrony in eggs stored for a longer time (*p* < 0.05). In conclusion, game quail eggs store well with little deterioration up to 28 days at 15.8 °C and 80% relative humidity, allowing for extended storage when shipping long-shelf-life eggs or assembling batches large enough to fully set an incubator in farms with small breeding flocks.

## 1. Introduction

The viability and quality of hatching eggs is a key aspect for the success of poultry production because it determines the quality of one-day-old chicks [1]. The viability of such hatching eggs depends on their proper handling during storage [2] and during incubation [1]. Incubation performance and the factors that affect it are well studied and known in the main poultry industry, such as laying hen breeders [3] and broiler breeders [4]. However, less and partial attention has been paid to the investigation of the factors that affect the incubation performance of the eggs of alternative poultry species, such as ostriches, partridges, quails, and others [5,6,7,8,9].

Among the alternative poultry species, quails of the genus *Coturnix* are disseminated in many countries of the world and are raised mainly to obtain meat [10] and eggs for human consumption [11] and produce birds for release in hunting preserves [12]. The quails to be released for hunting purposes are bred and raised on game farms in which the eggs laid by the female quail breeders are submitted to artificial incubation to obtain the chicks to be raised [13,14].

The handling of the hatching quail egg in the game farms before its incubation usually includes its storage for a certain time, between its laying and the setting of the next batch in the incubator [14,15]. Quail game farms with a breeding stock of a certain size and a large production usually set the incubators quite frequently at short intervals, even weekly [15], which allows the newly laid eggs to maintain good viability for their subsequent incubation. However, there are quail game farms that have small breeding stocks and reduced egg production, which need to store hatching eggs for longer periods while gathering a batch of sufficient size to fully set an incubator and subsequently obtain a batch of one-day-old chicks to be raised together, even if the eggs lose some viability [14]. Game farming of quails is carried out using the Japanese quail (*Coturnix japonica*), the European species (*Coturnix coturnix*), or hybrids of both species as breeding stock [12,13,14,15,16]. Farms that breed the two latter types of quail show a more seasonal pattern of egg production than farms that breed the Japanese species because of the marked reproductive seasonality of the European quail due to its wild nature [17]. For this reason, the time these quail game farms store the hatching eggs—at the beginning and at the end of the laying season when lower laying frequencies occur [14]—often exceeds the recommended time during which eggs do not lose their viability, which ranges from 7 to 15 days in game quail [13,14]. In addition, there are quail game farms that sell and ship hatching eggs [12], for which the shelf life should be as long as possible. However, the maximum duration of the storage period recommended for hatching game quail eggs (from 7 to 15 days) has been reported in informative publications that do not provide experimental evidence [13,14,15].

The maximum time for which it is recommended to store hatching eggs before incubation varies according to poultry species and production systems. In broiler and laying hen breeders’ eggs, it is recommended not to exceed storage periods of 7 days [18,19]. In ostriches, it has been reported that the eggs suffer a significant drop in viability after two weeks of storage [20]. However, there are avian species whose eggs resist longer periods of storage before incubation, without significant deterioration in their viability. Among these species, it is worth highlighting the partridges of the genus *Alectoris*. In fact, it has been reported that chukar partridge eggs (*Alectoris chukar*) [21] and red-legged partridge eggs (*Alectoris rufa*) [6] can be stored up to 28 days without significant deterioration in their hatching performance. In contrast, red-legged partridge eggs stored for 35 to 42 days are known to lose viability during incubation [6,8]. A study [22] has shown that hatchability of fertile eggs can be kept for seven weeks in rock partridge (*Alectoris graeca*). In Japanese quail, it has been reported that hatching eggs store well up to 7 days [23,24], while storing them beyond 7 days [25,26,27], 8 days [28,29], 10 days [30], 12 days [31], or 14 days [21,32], according to the experimental conditions, impairs their hatchability. Only one study has been found in the literature that investigates the effect of the storage time of European quail eggs under game farming conditions, which reports a reduction in the hatching rate of the eggs set between days 4 and 7 of storage [33].

In spite of the fact that there are many quail game farms distributed throughout several countries [12,34,35], there is a lack of scientific research concerning the hatching performance of farmed *C. coturnix* eggs submitted to long-term storage before incubation. In order to improve the long-term storage and shipment of hatching eggs by game farmers, reference values of the hatching performance of long-term stored game quail hatching eggs will have practical applications. In this context, this research aims to evaluate the effect of 0-, 7-, 14-, 21-, 28-, and 35-day storage periods on weight loss during the storage and incubation periods, chick weight at hatch, hatchability, and the incubation length of game quail eggs.

## 2. Materials and Methods

### 2.1. Breeding Flock and Egg Sampling

The trial was carried out using hatching eggs gathered during September and October 2020 from a small commercial game quail (*Coturnix coturnix*; unverified genetic purity) farm (breeding flock: 80 females and 20 males) located in Bienvenida (province of Badajoz), southwest Spain (Latitude: 38°16′22.17′′ N; Longitude: 6°12′10.25′′ W). The breeding quails were kept in groups of 10 birds, formed by two males and eight females in flat-deck cages measuring 0.38 × 0.75 × 0.36 m (width × length × height) per department. The birds were housed inside a building with natural ventilation and natural lighting (varying from 12L:12D to 10.7L:13.3D), without photoperiod supplementation. Breeding quails were fed a commercial balanced feed in pelleted form (16.0% CP, 4.0% EE, 4.7% CF, 5.9% ash, 0.82% lysine, 0.27% methionine, 1.2% Ca, 0.61% P, 0.12% Na, and 3193 kcal ME/kg; Inalsa^®^, Torralba de Calatrava, Spain). The average live weight was 139 g for males and 176 g for females. All the eggs used in the trial were gathered from quails aged between 13 and 17 weeks, whose laying rate at the beginning of the egg collection was approximately 61%.

### 2.2. Experimental Design and Egg Storage Management

Two hundred and seventy-nine eggs were arranged in six experimental batches submitted to storage periods of different lengths before incubation: 0 days (*n* = 49 eggs), 7 days (*n* = 47 eggs), 14 days (*n* = 47 eggs), 21 days (*n* = 49 eggs), 28 days (*n* = 43 eggs), and 35 days (*n* = 44 eggs). This small sample size was due to the fact that this was the maximum number of eggs available because the farm had a small breeding flock. The eggs of all treatments were laid on the respective collection dates and were immediately moved to the laboratory and placed in the storage chamber. In all experimental treatments, eggs were stored small end down at 15.8 °C and 80% RH in a storage chamber (Vinotek^®^, Liebherr, Biberach an der Riss, Germany), being turned 90° twice a day at regular 12 h intervals.

### 2.3. Egg Incubation

Once the storage period was over, all eggs were pre-warmed for 12 h at 25.3 °C and 36% RH, before incubation, by keeping them in the hatchery. All egg batches were set into the incubator on the same date (26 October 2020). The incubator (Masalles HS25^®^, Masalles, Ripollet, Spain) was set at 37.8 °C and 55% RH, and the eggs were automatically turned 90° every hour. On day 14 of incubation, the eggs were transferred to a hatcher (Maino Incubators 2-630 XHM^®^, Maino Enrico-Adriano S.n.c., Oltrona di San Mamette, Italy), which was set at 37.5 °C and 80% RH, without turning the eggs until hatching. In order to unequivocally identify which chick hatched from each egg, when transferring the eggs from the incubator to the hatcher, they were placed in individual mesh baskets located in the hatcher trays.

### 2.4. Data Recorded and Variables Calculated

The eggs were individually identified before storage by writing the corresponding numbers on the shell using an indelible marker. All eggs were individually weighed before storage, when set in the incubator, and at day 14 of incubation. For each single egg, weight loss during storage was obtained as a percentage of egg weight at the beginning of the storage period. Egg weight loss during the first 14 days of incubation was obtained as a percentage of egg weight at the beginning incubation, and total egg weight loss was calculated as a percentage of egg weight loss between the beginning of the storage period and day 14 of incubation. The number of hatched chicks and the number of unhatched eggs were recorded once the incubation period finished, and the hatchability of the incubated eggs was calculated as the percentage of incubated eggs that hatched. All hatched chicks were weighed at hatch. Relative chick weight was calculated as the percentage of chick weight at hatch compared to the egg weight before incubation. Unhatched eggs were broken and examined macroscopically to determine true fertility [36]. Fertility was calculated as the percentage of incubated eggs that were fertile. The hatchability of the fertile eggs was calculated as the percentage of fertile eggs that hatched. Fertile eggs were classified into the following categories [8,36,37,38]: fertile without development (blastodisc not developed and with deteriorated outline); positive development (blastoderm developed but with an absence of blood formation); early embryonic mortality (up to stage 18—or day 3 of development—as described for quails by Ainsworth et al. [38]); late embryonic mortality (from stage 19 according to Ainsworth et al. [38]); pipped but chick not out of shell. Incubation length was obtained by means of hatching controls performed at 12 h intervals starting at day 15 of incubation as the difference between the hatching date (when the chick comes out of the shell) and the date the eggs were set in the incubator.

### 2.5. Statistical Analyses

Differences in fertility, the hatchability of incubated eggs, and the hatchability of fertile eggs among batches that were submitted to different storage periods before incubation were analyzed by means of contingency tables on which Pearson’s χ^2^ tests were performed. When significant values were found, the standardized residuals (*R*) were calculated, with *R* = 1.96 being considered the discriminant value for a confidence level of 95% [39]. Fertile eggs weights before storage, before incubation, and at 14 days of incubation; weight losses of fertile eggs during storage and during incubation; total weight loss (storage and incubation periods); chick weight and relative chick weight at hatch; incubation length; and embryonic mortality at each development stage of fertile eggs were analyzed as dependent variables by means of one-way analysis of variance, with storage length as a factor. Initial egg weight was considered as a covariate when analyzing subsequent egg weight and egg weight losses. When significant differences were found by the one-way analysis of variance, means were separated by means of Duncan’s tests [40]. All quantitative results are expressed as mean and standard error (SE). The descriptive statistics, including minimum, maximum, coefficient of variation, variance, kurtosis (g_2_), and skewness (g_1_) coefficients, were also calculated for the incubation length. Differences in the variance of the incubation length as a function of the egg storage length were also analyzed [40]. For all comparisons, statistical significance was accepted when *p* < 0.05. SPSS v. 15.0 software (SPSS Inc., Chicago, IL, USA) [41] was used to perform the statistical analyses.

## 3. Results

### 3.1. Fertility and Hatchability

Game quail eggs showed an average fertility of 83.2%, which did not vary between the treatments established according to the storage time of the eggs before incubation (*p* > 0.05; Table 1). The average hatchability of the incubated eggs was 54.8%, and the average hatchability of the fertile eggs was 65.9% (Table 1). Both hatchability parameters were affected by the storage time of the eggs before incubation so that eggs stored for 35 days had a lower hatchability than those stored up to 28 days (Table 1; *p* < 0.05). The drop in hatchability between the fourth and fifth week of storage was 50% (Table 1).

### 3.2. Embryonic Mortality

Total embryonic mortality of fertile game quail eggs accounted for 34.05% (Table 2) and was higher in eggs submitted to 35 days of storage before incubation (*p* < 0.05). The length of the storage period to which the eggs were submitted before incubation did not influence (*p* > 0.05) the stages of embryonic development at which mortality occurred.

### 3.3. Egg Weight and Egg Weight Loss during Storage and Incubation

The initial average weight of the game quail eggs was 10.20 g and showed differences (*p* < 0.05) between batches subjected to different storage periods before incubation (Table 3). Thus, the heaviest eggs were those from the batch that was not stored and the batch that was stored for 7 days, while the lightest were those from the batches stored for 21 and 28 days.

The average weight of the game quail eggs after the storage period, coinciding with their setting in the incubator, was 10.02 g (Table 3). These showed differences according to storage time (*p* < 0.05) so that the heaviest eggs were those stored up to 14 days, and the lightest were those stored for at least 21 days. The average weight loss of eggs during storage was 1.85% (Table 3), increasing progressively from 0 to 3.79% as the storage time increased (*p* < 0.05).

The average weight of the game quail eggs at 14 days of incubation was 8.91 g (Table 3). Differences were observed according to the batches subjected to different storage periods (*p* < 0.05) so that the heaviest eggs were those stored up to 7 days, and the lightest were those stored for 21 days. The average weight loss of the eggs during incubation was 11.17%, with eggs stored for 21 days showing the greatest weight loss and eggs not stored and eggs stored for 28 days before incubation showing the lowest weight loss.

The average weight loss of the eggs, considering the storage and incubation periods together, was 12.79% (Table 3). A progressive increase in total weight loss of the eggs was observed as the storage period increased (*p* < 0.05), varying between 10.18% in eggs that were not stored and 15.18% in eggs stored for 35 days.

### 3.4. Chick Weight at Hatch

One-day-old game quail chicks weighed 7.44 g on average at hatch and showed a relative weight of 72.48% (Table 4). All chicks hatched were normal, healthy, and well developed. The storage time before the incubation of the eggs from which the chicks hatched influenced their weight at hatch and their relative weight (*p* < 0.05) so that the heaviest chicks came from eggs stored for up to 14 days (7.60–7.66 g), and the lightest came from eggs stored for 35 days (7.04 g), and the relative chick weight was lower for chicks hatched from eggs stored for 35 days (69.47%).

### 3.5. Incubation Length and Hatching Synchrony

The average length of the incubation period was 16.68 days (Table 5) and was affected by the storage time of the eggs before incubation (*p* < 0.05). Thus, the incubation period increased progressively as the storage time of the eggs increased, varying between 16.47 days in eggs that were not stored and 17.86 days in eggs stored for 35 days.

The length of the incubation period also showed differences in its dispersion according to the length of the storage period before incubation (Table 5). Thus, there were differences in the variance of the storage period (*p* < 0.05), with eggs stored for 21 and 35 days showing greater variances (0.108 and 0.105, respectively) and coefficients of variation (1.97 and 1.81%, respectively). Both treatments showed a skewed leptokurtic distribution of the incubation length: in eggs stored for 21 days, it was very likely skewed positively, while in eggs stored for 35 days, it was very likely skewed negatively. In the same vein, eggs stored for 21 days showed a greater hatching window, with chick hatching spread over 72 h. Conversely, eggs stored for 7 days before incubation displayed the greatest hatching synchrony as all chicks hatched at 16.5 days of incubation.

## 4. Discussion

To the best of our knowledge, this is the first study to investigate the effects of the long-term storage of game quail hatching eggs on incubation performance. In fact, the effects of storing European quail hatching eggs under game farming conditions have only been previously investigated for short storage periods of one week [33], whereas in Japanese quail, the effects of storing eggs up to 14–15 days have been extensively investigated [26,28,29,30,31,32]; only one study has tested egg storage up to 28 days before incubation in this species [21].

The average fertility of the eggs found in this research (83.2%; Table 1) was higher than that reported for *Coturnix coturnix* under game farming conditions (61.2%; [42]). Fertility in game farms that raise hybrid breeding quails has been reported to range between 65 and 80% [14]. This difference in fertility could be due to differences in the selection among breeding flocks [43]. The true fertility recorded by us for game quail was within the range of values described for domestic Japanese quail (68–97%; [29,31,32,44]). The good fertility of eggs from game quails in this trial might be due to the good condition of the breeding flock in the farm where the eggs were gathered. In fact, the true fertility of the eggs depends directly on the adequate fertilization of the eggs when the breeder quails mate and, therefore, on the factors that influence their reproductive efficiency, such as breeders’ age and strain or the quality of male and female gametes [45]. Moreover, the absence of differences (*p* > 0.05) in fertility between the different batches established according to the storage time before incubation (Table 1) indicates that eggs were laid during a period in which the possible effect of reproductive seasonality was not an obstacle to achieving optimal reproductive performance. This, together with other obtained results, suggests that the genetic purity of breeder quails used for this trial should be questioned, so it is probable that this flock may have a certain degree of crossbreeding, as previously cited for many other game farms [12,13,14,15,16].

The average hatchability of the set eggs (54.8%; Table 1) was within the range of values found by Caballero de la Calle et al. [33] in European quails under game farming conditions (50.5 to 68.9%). No studies have reported on the hatchability of the fertile eggs in game-farm-reared quails of the *C. coturnix* species. The hatchability of the set eggs and the hatchability of the fertile eggs (65.9%; Table 1) observed in the present trial for game quails were, however, below the interval of values described in the literature for Japanese quail eggs (82.4–91.5% for the hatchability of the set eggs and 91.2–95.8% for the hatchability of the fertile eggs [24,29]). Differences in hatchability between the game quails in this trial and values recorded in the literature for the Japanese quail might be due to the wild nature of the European quail, which leads to poorer reproductive performance in captivity [17,42].

The hatchability of all incubated eggs and that of the fertile eggs were significantly reduced when storage time before incubation was extended from four to five weeks (Table 1; *p* < 0.05). This is a relevant finding of this research since, to date, the effects of the long-term storage of hatching quail eggs have never been investigated. In fact, it has only been reported that hatchability is reduced by extending the storage time to 7 days in European quail eggs [33] and between 7 and 14 days in Japanese quail eggs [21,25,26,27,28,29,30,31,32]. Our finding agrees with previous reports on the long-term storage of chukar partridge [21] and red-legged partridge eggs [6], which also maintain high hatchability when stored for up to 28 days before incubation. In the same vein, the hatchability of fertile eggs can be maintained for seven weeks in rock partridge (*Alectoris graeca*) [22]. As reported in the literature, as the egg age increases, it loses CO_2_ and moisture through shell pores, the albumen and yolk pH increase, and the albumen height and vitelline strength decrease. The CO_2_ loss must come at the right time and an excessive carbon dioxide loss during egg storage leads to a high albumen pH, which may have a negative effect upon the initiation of embryonic growth and its subsequent viability [18].

The increased total embryonic mortality observed in this trial following the extension of the pre-incubation storage period up to 35 days agrees with other results on total embryonic mortality reported by Hamid and Uddin [46] in chicken, by Gómez-de-Travecedo et al. [8] in red-legged partridge, and the tendency observed by Seker et al. [31] in Japanese quail when the storage period was extended beyond the recommended duration. In our trial, this might be due to the cumulative effects of a tendency to increased late embryonic mortality and higher (although not significant) proportions of positive development and late embryonic mortality when the eggs were stored for 35 days (Table 2).

The mean weight of recently laid game quail eggs obtained in this trial (10.2 g; Table 3) was in the upper level of the range (8–10 g) described for game quail under farming conditions by Caballero de la Calle et al. [47]. It was slightly higher than the average weight of pure European quail eggs (9.3 g) and coincided with the average weight of F2 hybrid quail eggs (10.2 g) reported by Caballero de la Calle et al. [17] in experiments under game farming conditions. As expected, the average weight of game quail eggs found in this trial was lower than the Japanese quail egg weights reported in the literature (10.2–13.3 g [24,29,30,44,48]), which vary depending on the genetics of the breeding stock. Storage treatments significantly influenced (*p* < 0.05) the percentage weight loss of fertile eggs during storage. Thus, the eggs stored for 35 days lost 1.8% of their initial weight, while those stored for 7 days lost only 0.86%. The progressive variation of egg weight loss during long-term storage, depending on the storage duration, fitted the pattern described by Tilki and Saatci [49] for rock partridge eggs stored at 15–18 °C and 70% RH for 7 to 35 days, by González-Redondo [6] for red-legged partridge eggs stored at 15 °C and 80% RH for 7 to 35 days, and by Gómez-de-Travecedo et al. [8] for red-legged partridge eggs stored at 15 °C and 80% RH for up to 42 days. When storage duration is extended, a deterioration of the internal quality of the fertile egg occurs due to the increasing loss of moisture [49,50,51]. This occurs through the pores of the eggshell, and the rate of the loss is influenced by the temperature and relative humidity in the storage chamber so that, if water loss is too low or too high, embryo development is affected [30].

The average weight loss of the fertile eggs during the first 14 days of incubation was 11.17% of the egg weight before incubation (Table 3). This value coincided with the value reported by Gómez-de-Travecedo et al. [8] for red-legged partridge eggs stored for 42 days at 15 °C and was slightly higher than the values found by González-Redondo [6] for red-legged partridge eggs stored for periods ranging between 0 and 35 days at 15 °C, although it represented a lower weight loss than that found for rock partridge eggs stored at 14 °C (14.6%) by Kırıkçı et al. [52]. However, our results do not match those of Romao et al. [30], who reported that Japanese quail eggs that are not stored presented an 8.3% weight loss, while those stored for 14 days showed a 4.4% weight loss during incubation. The weight loss percentages of fertile eggs during the first 14 days of incubation in our study (*p* > 0.05) did not follow a clear pattern as a function of storage time before incubation. However, total egg weight loss of the fertile eggs increased progressively for treatments of eggs stored during longer periods (particularly for eggs stored for 21 and 35 days; Table 3). Greatly increased total weight loss in batches of eggs stored for long periods was previously observed for rock partridge by Kirikçi et al. [52] and for red-legged partridge by González-Redondo [6] and Gómez-de-Travecedo et al. [8].

The weight loss observed in this trial for game quail eggs, which is probably involved in the progressive deterioration of the internal quality of the egg during storage, is responsible for the observed fall in hatchability and the early deterioration of the embryo development. This probably results from the increased evaporative losses of the eggs [31], which is particularly intense when eggs are stored for too long [6,8].

The chick weight at hatch (7.44 g; Table 4) recorded in the present trial coincides with what was previously described for F2 hybrid chicks obtained from the backcross of *Coturnix coturnix* and Japanese quail (7.22 g) and is slightly higher than that reported for purebred *Coturnix coturnix* chicks (6.85 g) in research carried out under game farming conditions; conversely, it was lower than the weight of Japanese quail chicks (9.89 g) or F1 hybrid chicks from the backcross of *Coturnix coturnix* and Japanese quail (8.21 g) [53]. In addition, for Japanese quail, chick weight at hatch has been reported to vary between 7.8 and 11.2 g [23,24,29,54]. The relative chick weight of game quail from this trial (72.5%; Table 4) was slightly higher than that reported for Japanese quail (68–71%; [25,54,55]), a difference that could probably be due to differences in the incubator and hatcher parameter settings among trials.

The chick weight at hatch was affected by the length of the storage period before the eggs’ incubation and showed a decrease as the storage period increased, particularly after 14 days of storage; e.g., chicks hatched from eggs stored for 35 days weighed 7.7% less than chicks hatched from eggs stored up to 14 days (Table 4). The literature shows that no correlation is found between storage length and chick body weight in Japanese quail eggs stored below 14 days [23,24,29,30,54]. Garip and Dere [48] did not find an effect of the storage length in quail eggs stored at 11 °C for up to 15 days, while eggs stored at 21 °C and 27 °C led to chicks whose weight at hatch decreased as the storage length increased, depending on the strain tested. In the same vein, previous research on broiler eggs storage has reported that chick weight at hatch declined with long storage periods [56,57,58]. Conversely, relative chick weight decreased in the present trial for eggs stored for 35 days (69.5%) compared to eggs stored for shorter periods (at least 72%; Table 4). In Japanese quail, no effect of the storage length on relative chick weight has been reported for storage periods lasting up to 10 days [25,54,55], although, in our trial, very long storage periods could have caused lower chick weights, given the well-known positive correlation between egg weight loss and chick weight loss when storage length is extended [27].

The average incubation length of game quail eggs recorded in this trial (16.68 days; Table 5) was in the lower level of the range found in the literature for the Japanese quail (390–460 h according to Mirosh and Becker [59], Petek et al. [23], Yildirim [60], and Schmidt et al. [61]). No reports have been found in the literature for this parameter in game quail farms. The incubation length was affected by the length of the storage period before incubation, with the eggs stored for 35 days hatching more than 24 h later than eggs stored up to 21 days (Table 5). Our results agree with reports confirming that longer storage periods delay hatching time in Japanese quail eggs when the storage period is extended up to 7 days [23], 8 days [54], or more than 13 days [59,60]. These results also follow the pattern of previous research on other species such as broiler, ducks (*Cairina moschata* and *Anas platyrhynchos domestica*), and red-legged partridge [2,8,58,62]. Long-term egg storage influences embryo development and metabolism, so embryos grow at a slower rate than short-term stored eggs, leading to longer incubation periods [2].

The hatching synchrony was worsened by prolonging the storage duration of game quail eggs before incubation, as revealed by the higher variance (*p* < 0.05) and coefficient of variation of the incubation length, particularly in the batches held for 21 and 35 days before incubation (Table 5). These results agree with what was reported by Özlü [63] in broiler breeders, for which prolonged egg storage led to an asynchronous hatching time. This worsening of the hatching synchrony could be due to the fact that the prolonged storage of the eggs before incubating them produces variations in the embryonic development of the organs, as well as in the embryo as a whole, which leads to a variation in the hatching window [60].

The ability of game quail fertile eggs to maintain high viability when they are submitted to long-term storage before incubation, which coincides with what was reported in partridges [6,8,21,22], could be partly explained by the evolutionary adaptation of these steppe species to lay large clutches that remain exposed in the nest to a warm and dry environment for many days before the female begins incubation [64].

Previous research on embryo development in avian species has demonstrated that, for optimal performance in hatching eggs, lower temperatures are advisable for longer storage periods, while higher temperatures are better for shorter storage periods [62,65,66]. Thus, further research could investigate the combinations of temperature and humidity during storage that best maintain the viability of game quail eggs in different scenarios of short, medium, and long-term storage before incubation.

## 5. Conclusions

In conclusion, game quail hatching eggs can be stored for up to four weeks without a significant loss of viability, which has been similarly reported in partridge eggs and contrasts the lesser durability of eggs from other poultry species. When eggs are stored beyond a month, hatchability and one-day-old chick weight are reduced, while incubation is prolonged, worsening hatching synchrony. Informative publications advise against storing game quail eggs more than 7 to 15 days [13,14] to avoid an unnecessary loss of hatchability. However, when longer storage periods are necessary for management or market reasons, it is possible to store game quail eggs during longer periods and still obtain an acceptable hatchability performance. This finding is relevant for the handling of quail hatching eggs in game farms, allowing a large enough batch of eggs to be gathered during prolonged periods to set in the incubator. It is particularly useful on small farms and also for the marketing and shipment of long-shelf-life hatching eggs as they may maintain optimal hatchability during storage.

## Figures and Tables

**Table 1 animals-13-02184-t001:** Fertility and hatchability of game quail eggs submitted to different storage periods before incubation.

Storage Length (d)	Number of Eggs			Fertility (%)	Hatchability of the Incubated Eggs (%)	Hatchability of the Fertile Eggs (%)
	Incubated	Fertile	Hatched			
0	49	42	31	85.7	63.3 ^a^	73.8 ^a^
7	47	38	30	80.9	63.8 ^a^	78.9 ^a^
14	47	44	31	93.6	66.0 ^a^	70.5 ^a^
21	49	41	27	83.7	55.1 ^a^	65.9 ^a^
28	43	34	23	79.1	53.5 ^a^	67.6 ^a^
35	44	33	11	75.0	25.0 ^b^	33.3 ^b^
Total	279	232	153	83.2	54.8	65.9
*p*-value				0.245	0.001	0.001

^a,b^ Values in the same column with different superscripts are different (*p* < 0.05).

**Table 2 animals-13-02184-t002:** Embryonic mortality of game quail fertile eggs submitted to different storage periods before incubation (mean ± SE).

Storage Length (d)	Fertile Eggs (n)	Fertile Without Development (%)	Positive Development (%)	Early Embryonic Mortality (%)	Late Embryonic Mortality (%)	Pipped but Chick Not Out of Shell (%)	Total Embryonic Mortality (%)
0	42	7.14 ± 4.022	9.52 ± 4.584	7.14 ± 4.022	2.38 ± 2.381	0.00 ± 0.000	26.19 ± 6.867 ^b^
7	38	5.26 ± 3.671	10.53 ± 15.045	2.63 ± 2.632	2.63 ± 2.632	0.00 ± 0.000	21.05 ± 6.702 ^b^
14	44	0.00 ± 0.000	9.09 ± 14.384	11.36 ± 4.840	9.91 ± 4.384	0.00 ± 0.000	29.55 ± 6.958 ^b^
21	41	0.00 ± 0.000	4.88 ± 3.406	17.07 ± 5.949	9.76 ± 4.692	2.44 ± 2.440	34.15 ± 7.498 ^b^
28	34	0.00 ± 0.000	8.82 ± 4.937	8.82 ± 4.937	14.71 ± 6.165	0.00 ± 0.000	32.35 ± 8.144 ^b^
35	33	3.03 ± 3.030	21.21 ± 7.227	24.24 ± 7.576	18.18 ± 6.818	0.00 ± 0.000	66.67 ± 8.333 ^a^
Total	232	2.59 ± 1.044	10.34 ± 2.004	11.64 ± 2.110	9.05 ± 1.888	0.43 ± 0.431	34.05 ± 3.118
*p*-value		0.174	0.338	0.067	0.116	0.459	0.001

^a,b^ Values in the same column with different superscripts are different (*p* < 0.05). SE: Standard error.

**Table 3 animals-13-02184-t003:** Egg weight and egg weight loss during storage and incubation of game quail fertile eggs submitted to different storage periods before incubation (mean ± SE).

Storage Length (d)	Number of Fertile Eggs	Egg Weight Before Storage (g)	Egg Weight After Storage (g)	Egg Weight Loss during Storage (%)	Egg Weight at 14 Days of Incubation (g)	Egg Weight Loss during Incubation (%)	Total Egg Weight Loss (%)
0	42	10.41 ± 0.093 ^a^	10.41 ± 0.093 ^a^	0.00 ± 0.000 ^e^	9.35 ± 0.127 ^a^	10.18 ± 0.946 ^b^	10.18 ± 0.946 ^c^
7	38	10.43 ± 0.140 ^a^	10.34 ± 0.139 ^a^	0.86 ± 0.086 ^d^	9.25 ± 0.144 ^a^	10.66 ± 0.404 ^ab^	11.41 ± 0.432 ^bc^
14	44	10.32 ± 0.132 ^ab^	10.16 ± 0.130 ^a^	1.49 ± 0.135 ^c^	8.96 ± 0.139 ^ab^	11.93 ± 0.621 ^ab^	13.21 ± 0.695 ^ab^
21	41	9.94 ± 0.150 ^b^	9.67 ± 0.149 ^b^	2.65 ± 0.213 ^b^	8.49 ± 0.153 ^c^	12.29 ± 0.559 ^a^	14.58 ± 0.697 ^a^
28	34	9.94 ± 0.143 ^b^	9.66 ± 0.142 ^b^	2.85 ± 0.155 ^b^	8.70 ± 0.136 ^bc^	9.98 ± 0.332 ^b^	12.53 ± 0.414 ^b^
35	33	10.12 ± 0.159 ^ab^	9.75 ± 0.163 ^b^	3.79 ± 0.193 ^a^	8.60 ± 0.175 ^bc^	11.87 ± 0.513 ^ab^	15.18 ± 0.623 ^a^
Total	232	10.20 ± 0.055	10.02 ± 0.059	1.85 ± 0.102	8.91 ± 0.063	11.17 ± 0.259	12.79 ± 0.299
*p*-value		0.020	<0.001	<0.001	<0.001	0.029	<0.001

^a–e^ Values in the same column with different superscripts are different (*p* < 0.05). SE: Standard error.

**Table 4 animals-13-02184-t004:** One-day-old game quail chick weight (g) and relative chick weight (%) according to different egg storage periods before incubation.

Storage Length (d)	Number of Chicks	Chick Weight at Hatching (Mean ± SE; g)	Relative Chick Weight (Mean ± SE; %)
0	31	7.66 ± 0.093 ^a^	73.55 ± 0.482 ^a^
7	30	7.60 ± 0.152 ^a^	72.50 ± 0.665 ^a^
14	31	7.63 ± 0.139 ^a^	73.01 ± 0.379 ^a^
21	27	7.10 ± 0.185 ^b^	72.00 ± 0.790 ^a^
28	23	7.30 ± 0.175 ^ab^	72.29 ± 0.586 ^a^
35	11	7.04 ± 0.248 ^b^	69.47 ± 0.761 ^b^
Total	153	7.44 ± 0.066	72.48 ± 0.259
*p*-value		0.022	0.010

^a,b^ Values in the same column with different superscripts are different (*p* < 0.05). SE: Standard error.

**Table 5 animals-13-02184-t005:** Incubation length of game quail eggs submitted to different storage periods before incubation.

Storage Length (d)	Number of Hatched Eggs	Incubation Length (d)						
		Mean ± SE	Variance	CV (%)	Skewness	Kurtosis	Minimum	Maximum
0	31	16.47 ± 0.040 ^d^	0.049 ^y^	1.34	−0.35	2.66	16.0	17.0
7	30	16.50 ± 0.000 ^cd^	0.000 ^z^	0.00	-	-	16.5	16.5
14	31	16.37 ± 0.040 ^d^	0.049 ^y^	1.36	−1.16	−0.70	16.0	16.5
21	27	16.63 ± 0.063 ^c^	0.108 ^x^	1.97	3.21	11.74	16.5	18.0
28	23	17.11 ± 0.044 ^b^	0.044 ^y^	1.23	1.47	0.16	17.0	17.5
35	11	17.86 ± 0.098 ^a^	0.105 ^x^	1.81	−2.42	5.51	17.0	18.0
Total	153	16.68 ± 0.038	0.216	2.79	1.51	2.23	16.0	18.0
*p*-value		<0.001						

^a,b,c,d^ Means in the same column with different superscripts are different (*p* < 0.05). ^x,y,z^ Variances in the same column with different superscripts are different (*p* < 0.05). SE: Standard error. CV: Coefficient of variation.

## Data Availability

Data are available from the corresponding author upon request.

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
