# Peer review of "Effects of Long-Term Storage on Hatchability and Incubation Length of Game Farmed Quail Eggs"

_animals, 2023, doi:10.3390/ani13132184_

Round 1

Reviewer 1 Report (Previous Reviewer 1)

Why did you use so few eggs in the experiment?

The hatchability of control group eggs is around 63%. Under normal conditions, hatchability should be 85%. Why is the hatchability of set eggs so low?

Change the word "load" to "set" throughout the article. It would be more appropriate to use the word "set" in incubation processes.

Line 120: Replace “n=43 days”  with “n=43 eggs”

Line 143: Replace “at incubator loading” “at the beginning incubation”.

Lines 195: delete the sentence “although a marginal tendency (p < 0.1) to higher early embryonic mortality was observed 195 as storage time before incubation increased”. Since you are analyzing at 0.05 significance level, it would not be appropriate to write 0.10 significance level.

Author Response

Answer to Reviewer 1 second round

General comments from the authors: we have accepted all the modifications proposed by the Reviewer, and we explain how we have addressed the modifications in the manuscript. We have evidenced in green color the parts modified in response to Reviewers’ requests, and typed in blue color several corrections done to improve English writing.

Reviewer’s comment: Why did you use so few eggs in the experiment?

Authors’ response: Although a larger sample size would be desirable, we cannot change it because the experiment has already been performed. We did not reach a higher sample size because we gathered the eggs from a commercial farm of small flock size, with low production, and only few surplus eggs were available to buy for our trial, and taking in account that at each sampling date the eggs used in the trial were laid in the sampling day. In fact, in Spain many of the quail game farms have breeding flock of low size. In any case, there are similar studies with a sample size like the one used by us (Romao et al., 2008; González-Redondo, 2010; Roriz et al., 2016, and more). Moreover, the results obtained have been statistically analyzed and interpreted consistently with the sample size. Anyhow, in Lines 104-105 of the corrected manuscript we indicated that the eggs were gathered from a small commercial game farm (breeding flock consisting of 80 females and 20 males), and in this manner the potential readers can understand the origin of the small sample size.

Reviewer’s comment: The hatchability of control group eggs is around 63%. Under normal conditions, hatchability should be 85%. Why is the hatchability of set eggs so low?

Authors’ response: Hatchability of the eggs set in the treatment not submitted to storage was 63.3% and, as you indicate, below the normal value of 85% for Japanese quails, because we have performed our study on game quails whose genetic basis is the European quail (Coturnix coturnix), which is known for its lower reproductive performance in captivity (lower than that of the Japanese quail). We have already discussed this point (Lines 284-293 in the original article; Lines 283-292 of the corrected version), as follows: “The average hatchability of the eggs set (54.8%; Table 1) was within the range of values found by Caballero de la Calle et al. [33] in European quails under game farming conditions (50.5 to 68.9%). No literature has been found reporting hatchability of the fertile eggs in game farm-reared quails of the C. coturnix species. Hatchability of the eggs set and hatchability of the fertile eggs (65.9%; Table 1) observed in the present trial for game quails were, however, below the interval of values described in the literature for Japanese quail eggs (82.4-91.5% for hatchability of the eggs set and 91.2-95.8% for hatchability of the fertile eggs [24,29]). Differences in hatchability between game quails in this trial and values recorded in the literature for the Japanese quail might be due to the wild nature of the European quail, which leads to poorer reproductive performance in captivity [17,42].

Reviewer’s comment: Change the word "load" to "set" throughout the article. It would be more appropriate to use the word "set" in incubation processes.

Authors’ response: We have accepted your recommendation and changed “load” to “set” in Lines 20, 34, 54, 56, 60, 128, 140, and 208 of the corrected manuscript.

Reviewer’s comment: Line 120: Replace “n=43 days”  with “n=43 eggs”

Authors’ response: We have changed “days” with “eggs”, as can be seen in Line 120 of the corrected manuscript. Sorry for the mistake.

Reviewer’s comment: Reviewer comment: Line 143: Replace “at incubator loading” “at the beginning incubation”.

Authors’ response: We have accepted your recommendation and have rewritten this part as “at the beginning incubation”, as can be seen in Line 143 of the corrected manuscript.

Reviewer’s comment: Lines 195: delete the sentence “although a marginal tendency (p < 0.1) to higher early embryonic mortality was observed 195 as storage time before incubation increased”. Since you are analyzing at 0.05 significance level, it would not be appropriate to write 0.10 significance level.

Authors’ response: We have accepted your recommendation and have deleted the sentence in Line 195 of the original manuscript: “although a marginal tendency (p < 0.1) to higher early embryonic mortality was observed as storage time before incubation increased”.

The other modifications that can be found in the manuscript have been done to satisfy requests done by the other Reviewers.

The authors thank Reviewer 1 for his/her useful comments, which have permitted us to improve our manuscript.

Reviewer 2 Report (Previous Reviewer 2)

accept after minor revision 

moderate editing of English langauge required   

Author Response

Answer to Reviewer 2 second round

General comments from the authors: we have accepted all the modifications proposed by the Reviewer, and we explain how we have addressed the modifications in the manuscript. We have evidenced in green color the parts modified in response to Reviewers’ requests, and typed in blue color several corrections done to improve English writing.

Reviewer’s comment: moderate editing of English language required.

Authors’ response: We have accepted your recommendation and have improved English writing. Improvements of English language have been marked in blue color in the corrected version of the manuscript. For example, see Lines 29, 57, 76, 102, 113, 119, 142, 185, 210, 272, 275, 281, 290, 316, 325, 327, 357, 364, 366, 368, 370, 376, 379, 407 and more.

The other modifications that can be found in the manuscript have been done to satisfy requests done by the other Reviewers.

The authors thank Reviewer 2 for his/her useful comments, which have permitted us to improve our manuscript.

Reviewer 3 Report (Previous Reviewer 3)

Comments attached.

Some tenses need changing.

Author Response

Answer to Reviewer 3 second round

General comments from the authors: we have accepted all the modifications proposed by the Reviewer, and we explain how we have addressed the modifications in the manuscript. We have evidenced in green color the parts modified in response to Reviewers’ requests, and typed in blue color several corrections done to improve English writing.

Reviewer’s comment: English: Some tenses need changing

Authors’ response: We have accepted your recommendation and have improved English writing, including revision of tenses. Improvements of English language have been marked in blue color in the corrected version of the manuscript. See, for example, Lines, 29, 57, 76, 102, 113, 119, 142, 185, 210, 272, 275, 281, 290, 316, 325, 327, 357, 364, 366, 368, 370, 374, 376, 377, 379, 407 and more

Reviewer’s comment: report actual p-values

Authors’ response: In the corrected manuscript resubmitted we have indicated p-value levels (p<0.05 or p>0.05) within the text under request of other of the Reviewers. Moreover, actual p-values are already indicated in the Tables. Therefore, we prefer not to replace in the text p-value levels with actual p-values, because actual p-values can be seen in Tables while p-value levels can be seen within the text.

Reviewer’s comment: Line 71, what was their reported duration? [In reference to maximum duration of the egg storage before incubation]

Authors’ response: The recommended time in informative publications without experimental evidence [13-15], during which eggs do not lose their viability, ranges from 7 to 15 d in game quail [13,14], as we have already indicated in Lines 66-69 of the original manuscript: “For this reason, the time these quail game farms store the hatching eggs at the beginning and at the end of the laying season, when lower laying frequencies occur [14], often exceeds the recommended time during which eggs do not lose their viability, which ranges from 7 to 15 d in game quail [13,14].” To gain clarity for the potential readers, we have indicated this duration (7 to 15 days” also in Line 72 of the corrected manuscript.

Reviewer’s comment: Table 2 Out of Shell

Authors’ response: We have corrected it as follows: “Pipped but Chick not out of Shell (%)”, because the style of this journal for the headings of the tables imposes writing the names of the variables with the initials in capital letter.

Reviewer’s comment: Correction in Table 3: a-e superscript

Authors’ response: In Table 3 we have corrected the mistake and have put “a-e” superscript in the footnote.

Reviewer’s comment: Line 303: Did you measure these parameters? If not, why not?

Authors’ response: We did not. The reason is because the storage chamber used (Vinotek®, Liebherr, Biberach an der Riss, Germany) is a commercial one that is not prepared to measure these parameters. However, to gain clarity for the potential readers, we have added “As reported in the literature,” at the beginning of this sentence (Lines 302 of the corrected version), thus permitting to understand that it refers to knowledge based on the literature and not directly derived from the results of our trial.

Reviewer’s comment: Line 323-324 “variables”

Authors’ response: This word does not refer to "variables", but in that sentence we mean that the weights of the eggs are variable according to the quail genotypes. To improve the clarity of the sentence, we have changed "are variable" to "vary" in the improved version of the manuscript (Line 323).

Reviewer’s comment: Line 374-375 ”led”

Authors’ response: We have corrected “led” instead of “lead” as can be seen in Line 374 of the corrected manuscript.

Reviewer’s comment: Line 377 declined

Authors’ response: We have corrected “declined” instead of “declines” as can be seen in Line 377 of the corrected manuscript.

The other modifications that can be found in the manuscript have been done to satisfy requests done by the other Reviewers.

The authors thank Reviewer 3 for his/her useful comments, which have permitted us to improve our manuscript.

Reviewer 4 Report (Previous Reviewer 4)

How it is possible to mark significant differences by superscripts (table 1 - hatchability; table 2 – total embryonic mortality) if, according to M&M the non-parametric test (ê­“2) was used to analyse these traits? Was the experiment replicated among particular groups?

Author Response

Answer to Reviewer 4 second round

General comments from the authors: we have accepted all the modifications proposed by the Reviewer, and we explain how we have addressed the modifications in the manuscript. We have evidenced in green color the parts modified in response to Reviewers’ requests, and typed in blue color several corrections done to improve English writing.

Reviewer’s comment: How it is possible to mark significant differences by superscripts (table 1 - hatchability; table 2 – total embryonic mortality) if, according to M&M the non-parametric test (ê­“2) was used to analyse these traits? Was the experiment replicated among particular groups?

Authors’ response: No, the experiment was not divided into replication groups. Fertility and hatchability (Table 1) were directly calculated as a percentage of the egg sample (as indicated in Lines 146-147 and 150-152 of the corrected manuscript). Thus, each single egg was considered a replicate. When performing chi-square test in the contingency tables for these variables, differences between treatments were evaluated throughout the standardized residuals (R), with R = 1.96 being considered the discriminant value for a confidence level of 95%, as was already indicated in Lines 164-166 of the original manuscript, and thus permitting us to mark significant differences by superscripts. In Table 2, embryonic mortality causes at each development stage of fertile eggs (assigning 100 value to eggs with this mortality cause and 0 with not mortality) were analyzed as dependent variables by means of one-way analysis of variance (see Lines 169-171 of the corrected manuscript), followed by Duncan’s tests to separate means when the anova displayed significant differences (see Lines 173-174 of the corrected manuscript).

The other modifications that can be found in the manuscript have been done to satisfy requests done by the other Reviewers.

The authors thank Reviewer 4 for his/her useful comments, which have permitted us to improve our manuscript.

This manuscript is a resubmission of an earlier submission. The following is a list of the peer review reports and author responses from that submission.

Round 1

Reviewer 1 Report

The number of eggs used in the study is very small. For hatching characteristics to be calculated as a percentage, each group must have at least 100 eggs.  In practice, storage is not done for such a long time. This is not suitable for the poultry sector. This app may only be suitable for small family poultry production.

Materials and Methods

Lines 108-109: In the study, it was emphasized that there is natural lighting. Light is an important factor in egg production. How many hours of lighting per day should be specified.

Lines 109-11: In terms of the protein value of the specified feed, it is more similar to laying hen feed. The protein value of quail feed should be at least 20%. In addition, it would be more accurate to give the energy value.

Line 112: Age is given on a weekly basis in poultry production.

Lines 114-117: This number of eggs is insufficient for storage and hatching results. In percentage calculations, there should be 80–100 eggs for each storage period.

Line 118: The temperature value varies in short- and long-term storage of hatching eggs. It would be appropriate to refer to the temperature and humidity values.

Lines 119: Is it mandatory to turn the eggs during storage?

Lines 131: Have your eggs been weighed individually? Are the weight values written on the egg? It would be better to write it more clearly.

Lines 138: Should it be clear whether the chicks are transferred to the individual hatches? This is important for individual weighing.

Lines 145-148: In the classification of embryonic deaths, it would be more appropriate to give it in days. The following resource may be useful in the classification of embryonic deaths in quails “https://doi.org/10.3382/ps.2011-01944

Lines 146-148: It is not appropriate to use "abortion" in poultry embryo deaths. It would be more appropriate to use embryonic mortality instead. 

Line 151: “loading date”? Please use the zootechnical term.

Statistical analyses

There are only two hypotheses in statistics, they are H0 and H1. In the null hypothesis (H0), the distribution is gaussian, the treatment effect is not significant. In the H1 hypothesis, the situation is the opposite. In the natural, health, and engineering sciences, the acceptance probability of the null hypothesis spreads up to 0.95/1. If a P value of 0.049 is found in a study with a significance level of 0.05, H0 is rejected. But this does not mean that the "more significant" difference occurs when a P value of 0.01 or 0.001 is found in the same study. There is no H2 or H3 hypotheses in statistics science. Therefore, use only one of these levels (0.05, 0.01, 0.001) in your studies.

Results

Table 2: “ a,b,c Values in”,  delete “c”

Table 4: “a,b,c,d Values in the same column”,  delete “c,d”

Conclusions

Line 402: “load”?

Reviewer 3 Report

Great! A few minor comments attached.

Reviewer 4 Report

Generally, the manuscript represents satisfying quality because the hen eggs are typically used as experimental material, also they are mentioned in the legislation. So, the use of quail eggs is quite original, however, some issues should be considered.

Line 71 – 72 – can authors point at papers recommending the storage duration of quail eggs?

M&M

What was the scale for embrionic developement used?

The groups of eggs were additionally divided into replication groups? If not how it was possible to evaluate the differences using post hoc test (table 1)?

Table 2 – rather the „early/late mortality” should be used instead of „early/late abortion”. „pipped but not out of shell” means internal pipping? Or pipped but unhatched chicks?

Table 5 – is it possible to show the hatching window in hrs instead of days?

Line 274 – although the birds in the manuscript as named as „game birds” their rearing method is typical for the farm (cage system). So, it is not true that there is no data about hatchability of quails. Some examples are mentioned below:

·         https://doi.org/10.3390/ani10020264

·         Soliman, F.N.K.; Elsebai, A.; Abaza, M. Hatchability traits of different colored Japanese quail eggs in relation to egg quality and female blood constituents. Egypt. Poult. Sci. J. 2000, 2, 417–430

·         Uddin, M.S.; Paul, D.C.; Huque, Q.M.E. Effect of egg weight and pre-incubation holding periods on hatchability of the Japanese quail eggs in different seasons. Asian-Austral. J. Anim. 1994, 7, 499–503.

·         https://doi.org/10.1016/j.psj.2022.102066

and many others.

It is impossible to verify some references listed in the manuscript, also due to the language of publication (i.e. 13, 14, 16 etc.)

The English language is comprehensible, most of it using vocabulary appropriate to the scientific field.
